



# ESD Ideas: Long-period tidal forcing in geophysics — application to ENSO, QBO, and Chandler wobble

Paul R. Pukite[1]

[1]DAINA, Minneapolis, MN, 55421, USA

*Correspondence to*: puk@umn.edu

**Abstract.**

*Apart from its known impact to variations in the Earth's length-of-day (LOD) variations, the role of long-period tidal forcing cycles in geophysical behaviours has remained relatively unexplored. To extend this idea, tidal forcing is considered as a causative mechanisms to the following cyclic processes: El Niño Southern Oscillation*

*(ENSO), Quasi-Biennial Oscillation (QBO), and the Chandler wobble. Subtle mathematical insights are required to make the connection to the observed patterns as the underlying periods are not strictly commensurate in relation to harmonics of the tidal cycles.*

There are three cyclic perturbations in the Earth's behavior that scientists have had difficulty pinning down. The actual understanding is so poor that there is no clear consensus for any of the behaviours, and the actual mechanism in each is

considered an as-yet unresolved mystery. One behavior has to do with an oceanic cycle, one with an atmospheric cycle, and one with the solid Earth. A consensus agreement is lacking in each of these three behaviours in spite of the fact that there may be an obvious yet mathematically-challenging common-mode cause tying them together. The challenge lies in simplifying the math of fluid dynamics and applying the appropriate signal processing techniques. With that, an elegant analytical framework can be applied to perhaps solve the mystery once and for all.

1. **Oceanic cycle**: This is the *El Niño Southern Oscillation* (ENSO) which is an erratic cycling of ocean surface temperature along the equatorial Pacific Ocean (McPhaden *et al*., 2006). Via a solution to the equations of fluid dynamics, derived from Navier-Stokes as the simplified Laplace's Tidal Equations as described in Pukite *et al*. (2018), it is straightforward to model the ENSO behavior. The key is to realize that the lunar tidal period and the annual cycle provide the forcing, thus periodically sloshing the ocean's thermocline upwards to the surface to produce the oscillating temperature behavior.

It is only difficult to interpret because the fluid dynamics is highly nonlinear, and thus a behavior that was deviously concealed from a routine conventional tidal analysis becomes more readily amenable to a specially structured kind of tidal harmonic modeling, calibrated to well-characterized angular momentum changes such as the Earth's length-of-day (LOD) variations. Although the ENSO dipole is the main focus of the model, it appears to be applicable to the other oceanic indices, such as IOD, NAO, etc.





2. **Atmospheric cycle**: This is the *Quasi-Biennial Oscillation* (QBO) of equatorial stratospheric winds. The QBO cycle is with respect to the east-west direction of the prevailing wind, which reverses on average every 14 months, and thus having a repeat period of ~28 months. Again, like ENSO, the key to modeling is to consider the interaction of the lunar nodal tidal cycle with the annual nodal cycle (see **Figure 1**), which together serve as the sensitive gravitational trigger to reversing the wind direction. Why this straightforward interpretation was missed is a mystery, but it likely has a genesis

in the work of R. Lindzen, who proposed a working hypothesis early on (Lindzen, 1987) that served more as a reference point than a final answer, as can be deduced from the more than 30 differing results of CMIP6 model runs (Richter et al., 2020). As it happens, the lunisolar long-period tides are consistent with the suggestion of gravity waves as a major contributing forcing mechanism – and although Lindzen dismissed known tidal period (Lindzen, 1967; Lindzen *et al*., 1974), he apparently never considered *aliasing* of the tidal cycles as a match to the 28 month cycle (Pukite *et al.*, 2018).

3. **Solid Earth cycle**: This is the *Chandler wobble* of the Earth's axis of rotation, revealed most clearly by a circular polar motion of several meters radius and a ~433 day period with respect to the solid Earth. Although typically attributed to internal core dynamics, it seems much more plausible and parsimonious to suggest again (as with the QBO, see **Figure 1**) an interaction of the lunar nodal and annual cycle supplying cyclic gravitational torque to the non-spherical Earth (note that a perfectly uniform sphere will experience no torque) (Pukite *et al*., 2018). So even though this aliased cycle will

match precisely the 433 day period, it's a mystery why this mechanism had not been previously identified. The likely rationale is that since Euler (in 1765) (Gross, 2000) had predicted a natural resonant period of 305 days based on the Earth's ellipticity, then the successive theories had assumed this would be modified slightly upwards due to the fluid nature of the Earth's core and oceans.

What all three proposed models have in common is a premise based on a ***forced*** response to external lunar and solar stimuli,

and not a ***naturally resonant*** response to non-specific sources of energy as seems to be prevalent as the more commonly-held views reported in the scientific literature. This may stem from a physicist's desire to find *eigenvalues* in nature to elegantly explain some novel behavior, as opposed to a more empirically derived *cause-and-effect* explanation first. As far as rethinking these mechanisms in a parsimonious fashion, the simplest explanation is perhaps the best explanation. It would indeed be rare to find phenomena as large as these three that would spontaneously occur without a straight-forward forcing mechanism.

The impetus to presenting this view is to motivate other geophysicists to evaluate the modeling framework against the available data sets. As with conventional tidal analysis, the fitted models improve over time as independent evaluations are carried out with incrementally advanced techniques. Details of these three models and the common lunar and solar mechanisms driving each are described in Pukite *et al.* (2018) – see chapter 11 for QBO, chapter 12 for ENSO, and chapter 13 for the Chandler wobble. A source code repository is available for evaluation and further exploration of these and other data sets such as LOD

and other climate indices at https://github.com/pukpr/GeoEnergyMath



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

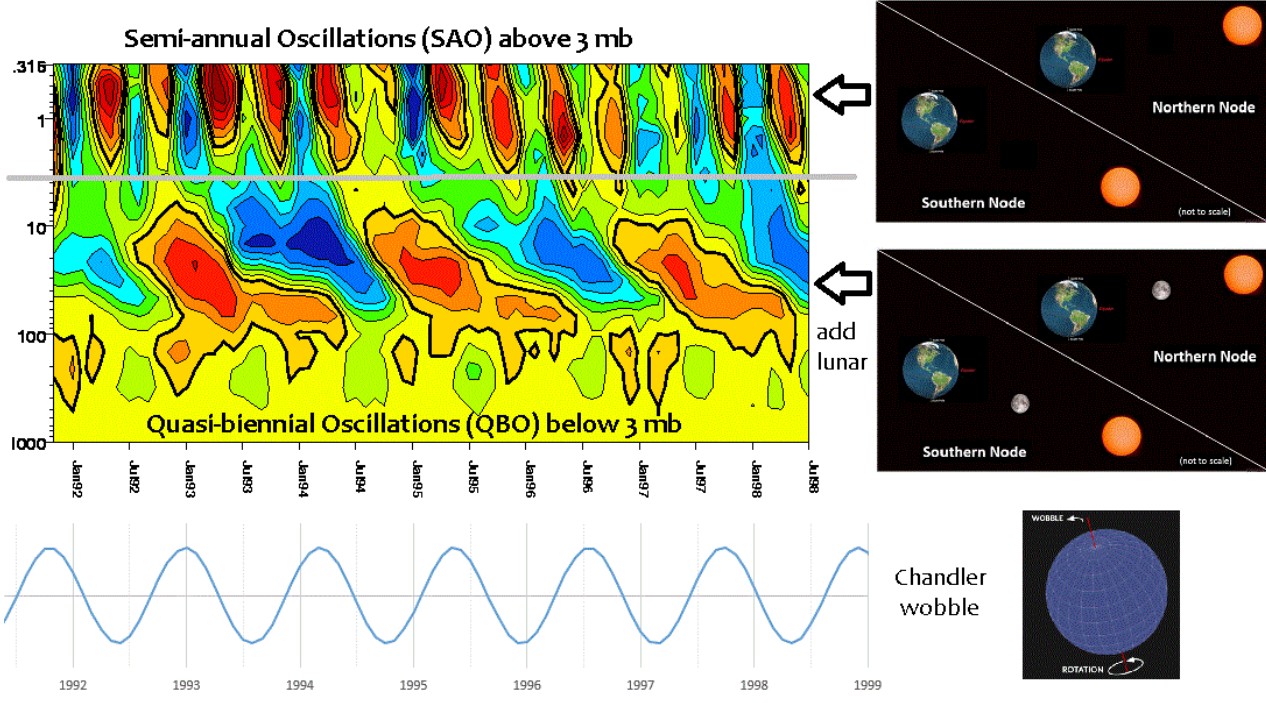

**Figure 1: The middle panel describes a mechanism for the QBO — the nodal cycle of the moon and sun trigger a reversal. The lower panel extends the concept to the Chandler wobble in the Earth's rotation axis due to a cyclic torque.**