# Peer review of "ESD Ideas: Long-period tidal forcing in geophysics — application to ENSO, QBO, and Chandler wobble"

_Earth System Dynamics, 2020_

## Referee Comment (RC1) · William Kessler (Referee) · 28 Nov 2020

I am a physical oceanographer who knows nothing about the Chandler wobble, is only slightly familiar with the QBO, but is a longtime expert on ENSO.

To be blunt, trying to shoehorn ENSO into a periodic tidal framework stretches reality to fit someone's preconceived theory. Only the most motivated reasoning can believe this.

Over the past four decades we have seen numerous out-of-nowhere theories - including those with apparent hindcast skill and much more plausible than the one here - attempt to explain and predict ENSO - but they have all failed at the only realistic test of such theories: a successful prediction. I have resolved never to pay attention to such

theories until they produce such a (public) prediction that subsequently comes true; that resolve has saved me from wasting lots of time. I commend this approach to readers of ESD. Let these authors announce an advance prediction of an El Nino event, then publish if and only if it verifies. Apparently, since the approach here is periodic, they should be able to produce a prediction of all the events over decades, past and future.

The best way to see the non-tidal nature of ENSO is to note that its behavior is well-represented in models of the coupled ocean-atmosphere system ranging from the idealized (e.g. Cane and Zebiak 1987 MWR) to modern GCMs ... none of which contain tides. These models DO have predictive skill, which is regularly tested by issuing detailed public forecasts. This is by no means to say that the problem is solved, but that the mechanisms involved are contained within the known (but very complex and multi-scale) nonlinear interactions of the ocean-atmosphere system. A voluminous literature digs into these interactions, explaining the coupled feedbacks that enable ENSO behavior and provide insight that is useful to understand less-dramatic signals in other tropical basins. Our understanding is NOT "so poor that there is no clear consensus for any of the behaviors" (L13-14). The authors should read the McPhaden et al paper they cite more carefully, and dig into its reference list that would correct this statement.

Various clues in the submission suggest the authors' lack of familiarity with the field: L10: "Subtle mathematical insights are required to make the connection". Apparently the authors' minds are more subtle than those of generations of geophysicists. L17-20: "The challenge lies in simplifying the math of fluid dynamics and applying the appropriate signal processing techniques. With that, an elegant analytical framework can be applied to perhaps solve the mystery once and for all." Well. Mathematical physicists have developed simplifications of Laplace's equations over 200 years. Many such are widely discussed. What the useful ones have in common is that they are physically-based; that is, they make scale approximations that isolate particular phenomena and thereby reveal their role in ENSO. One example would be the beautiful theory of equatorial planetary waves, central to ENSO. L23-24: "it is straightforward to model ENSO behavior". If it's so straightforward, please do this, and publish your prediction. You'll be famous. L25: "deviously concealed". Nuff said.

I am sorry to have wasted an hour on this.

Billy Kessler, NOAA/PMEL, Seattle

———————————————————

---

## Author Comment (AC1) · 29 Nov 2020

Thank you for the review.

I agree that a validation step is needed for this particular ENSO model to be accepted. The usual issue with validation against future events is that a sufficient time interval must elapse before sufficient statistics are gathered for the model to be substantiated (or debunked). And some models, such as those for astrophysics, are forecast for millions of years in the future, so the lack of a predictive validation step is acceptable for that scientific discipline. However, as an alternative approach, often one can employ cross-validation techniques such that one portion of the time-series can be used as a training interval and another interval can be used as a test or validation step.

[Figure]

A simple signal processing technique that was hinted at in the idea submission can be used to substantiate that deterministic characteristics exist within the ENSO time-series. The technique applies a Fourier spectrum analysis to reveal an underlying double-sideband suppressed carrier modulation in the signal. If the measured signal does have a modulation imparted by a carrier signal, then the frequency spectrum should show a mirror symmetry about the carrier frequency – for every frequency a pair of positive and negative side-lobes will appear. See Fig. 1

As the general assumption is that an annual seasonal barrier is responsible for trigger-ing an El Nino (or La Nina) episode, the annual frequency provides the carrier signal which is used as the mirror symmetry point in the Fourier spectra. By mirror folding the frequency beyond the 0.5/year point, this double-sideband modulation is observed (nothing like this has been reported in the literature, as far as I know). See Fig.2

As mentioned in the paper, a solution of Laplace's tidal equations (LTE) was derived elsewhere (see Pukite et al, Mathematical Geoenergy, Wiley/2018) and that was used to train a time-series over a fixed ENSO interval, from 1880 to 2016 of the NINO34 data set. More recent data post-2016 is then used as a cross-validation step. Long-period tidal forcing calibrated to length-of-day (LOD) data was used as the modulating stimulus and the annual barrier was applied as the "carrier" signal – these are shown in steps A through G below in Fig. 3.

The cross-validation takes place when the model is extended beyond 2016, assuming the cyclic tidal pattern will continue – see the chart Fig.4:

Certainly this is not considered a perfect "blind test", as one can rightly state that fits that did not work were discarded. Yet, in the spirit of open analysis, others can replicate the calculations or experiment via the instructions that are included via the supplemen-tal information uploaded to the Copernicus page.

From my understanding, tidal forcing is being included in recent GCM packages. There is a disconnection between many of the reduced models used to demonstrate some
aspect of ENSO behavior, such as Cane-Zebiak, and the full-blown GCMs. The LTE formulation I am working from may provide a useful approach that is simple but bridges to the more comprehensive GCM solution.

Thank you again for reviewing the idea. Eventually some machine learning model based on a neural net connectivity may find a similar pattern match, yet with little insight on how the net was physically constructed (see citation [1] below). So whether or not it is completely understood, I fully expect that a ML algorithm will be able to make the same connections and provide a predictive capability. In contrast, the model described in this idea was developed from first principles and one can see the sequence of steps from forcing to response.

[1] Pukite, Paul. "Nonlinear Differential Equations with external forcing." ICLR 2020 Workshop on Integration of Deep Neural Models and Differential Equations. 2020.

[Figure]

$$\underbrace{V_m \cos(\omega_m t)}_{\text{Message}} \times \underbrace{V_c \cos(\omega_c t)}_{\text{Carrier}} = \underbrace{\frac{V_m V_c}{2} \left[ \cos((\omega_m + \omega_c)\, t) + \cos((\omega_m - \omega_c)\, t) \right]}_{\text{Modulated Signal}}$$

[Figure]

**Fig. 1.** Mechanism of double-sideband suppressed carrier modulation (DSSCM)

[Figure]

**Fig. 2.** ENSO spectrum showing DSSCM

[Figure]

**Fig. 3.** ENSO tidal forcing model

NINO3.4 Index / weatherzone.com.au

cross-
validation

←training to 1880

Data (NINO34) —ENSO model

2000    2010    2020

**Fig. 4.** ENSO cross-validation of out-of-band data

---

## Referee Comment (RC2) · Anonymous Referee #2 · 29 Dec 2020

The problem of the connection between ENSO, QBO and Chandler wobble with lunisolar tides was posed by Prof. Nikolai Sidorenkov, Dr. Ian Wilson and Leonid Zotov long time ago. The novelty of this manuscript is unclear and questionable. Please, see some recent papers below:

Wilson, I.R.G., and Sidorenkov, N.S, 2019, A Luni-Solar Connection to Weather and Climate II: Extreme Perigean New/Full Moons and El Niño Events, The General Science Journal, Jan 2019, 7637. DOI: 10.13140/RG.2.2.20846.87362

Sidorenkov N.S. Synchronization of terrestrial processes with frequencies of the Earth-Moon-Sun system //Astronomical and Astrophysical Transactions (AApTr), 2017, Vol. 30, Issue 2, pp. 249-260, ISSN 1055-6796

N. S. Sidorenkov. Celestial Mechanical Causes of Weather and Climate Change ISSN 0001-4338, Izvestiya, Atmospheric and Oceanic Physics, 2016, Vol. 52, No. 7, pp. 667–682. © Pleiades Publishing, Ltd., 2016. DOI: 10.1134/S0001433816070094

Sidorenkov N.S., 2009. The interaction between Earth's rotation and geophysical processes. Weinheim. WILEY-VCH Verlag GmbH & Co. KGaA. 2009. 317 pp. ISBN: 978 – 3 – 527 – 40875 – 7

---

## Author Comment (AC2) · 29 Dec 2020

Indeed, I have referenced the work of Sidorenkov as a cited paper. However, that research neither debunks the submitted idea nor does it clearly articulate the actual physical mechanism at work. For the Chandler wobble, Sidorenkov asserts that this is the predicted value for the frequency:

1/1.0 - (1/18.61 + 1/8.85) = 1/1.20 cycles per year

18.61 years is the nodal declination variation cycle of the moon while 8.85 years is the perigean cycle. According to this expression, it puts the Chandler wobble period at 438 days, which is off the generally accepted value of 433 days. That has two strikes against it – it is not very precise and has no clear conceptual basis.

[Figure]

What I did was much more elementary than Sidorenkov's heuristic and can be understood from introductory physics. Consider a rod rotating about its axis with ends of an attractive nature (e.g. could be magnetic, which is easy to demonstrate in the lab). As a forcing, we introduce two objects that cycle incommensurately between the two ends (i.e. nodally in earth terms). Due to the law of conservation of angular momentum, the axis of rotation will eventually precess at a rate fundamentally related to these two rates as a forced response. Next, analogize that one of the objects is the sun at a rate of 1 full nodal cycle per year, and the other is the moon at a rate of 1 full nodal cycle every 27.2122 days (the nodal or draconic lunar cycle), then the expected maximum strength conjunction cycle can be calculated via the algebra of aliasing:

(365.242/2) / (27.2122/2) - integer( (365.242/2) / (27.2122/2))

which is 0.422 cycles per half-year or (365.242/2) /0.422 = 432.75 days

See Figure 1 in the submitted ESD Ideas paper for a schematic of the geometry, and Figure 1 attached for a set of calculated conjunction cycles showing the 433 day cycle.

Note that this is much more precisely aligned with the actual Chandler wobble period and because the earth is not a perfect sphere and thus has a moment of inertia, this predicted forced response cycle MUST exist in any measurements. The only question is it's strength. Since it is a forcing, it will never dissipate and any natural resonance of the earth's wobble (i.e. the original 305 day cycle predicted by Euler) may actually amplify it's strength over that bandpass regime.

The applicability of Sidorenkov's assertions to ENSO and QBO are not as relevant, as my formulation requires an analytical solution to Laplace's Tidal Equations along the equatorial topological boundary. Especially for ENSO, the tidal forcing synchronization only emerges if this is considered.

This is not to say that Sidorenkov's hypothesis provided no motivation, as his ideas were evaluated, as were the suggestions of Munk & Wunsch and Keeling & Whorf in

terms of oceanic tidal forcing, and Richard Lindzen in regards to QBO, who on separate occasions claimed that "For oscillations of tidal periods the nature of the forcing is clear" [1] and "it is unlikely that lunar periods could be produced by anything other than the lunar tidal potential" [2]. The approach as described clears up how the tidal forcing is synchronized, thus allowing the climate science community to re-evaluate Lindzen's early concerns.

References [1] RS Lindzen, Planetary waves on beta-planes, Monthly Weather Review, 1967 [2] RS Lindzen, S Hong, Effects of mean winds and horizontal temperature gradients on solar and lunar semidiurnal tides in the atmosphere, Journal of the Atmospheric Sciences, 1974

—————————————————————

[Figure]

**Fig. 1.** Conjunction cycle of draconic fortnightly and semi-annual periods – 10 cycles in ~4330 days

---

## Short Comment (SC1) · 5 Jan 2021

To my knowledge, the problem discussed is not so easy, as it could be imagined. Of cause, El Nino oscillation and QBO seems to be reproduced by GCMs, but predictions are still comprehensive, for example strong El Nino 2016 was unexpected. At the same time LOD started to decrease after 2016 and it was also unexpected. Very strong El Nino repeats at the beginning of the declining phase of LOD, what makes us to think about possible connection of these phenomena. But simple tides also do not explain what we observe. Chandler wobble frequency does not present anywhere else in Earth systems, thus we do not see any pike in the excitation spectra at 0.843 cpy, at the same time we know from theory, that Chandler frequency is somewhere near resonance. Amplitude of Chandler wobble is changing with 40 and 80-year modulations.

[Figure]

In the combinations of solar and lunar frequencies we can find some close-bye modes. The interests of N. Sidorenkov, for sure, are in the field of interrelations of frequencies of El Nino, QBO and Chandler wobble. And in his book and papers this question was risen decades ago. I did not quite understood the author's calculations, why 0.42 – the remainder of the division of solar year by draconic month should be doubled to satisfy 0.84 cpy frequency of Chandler wobble, but I know that in the publications of Sidorenkov it is mentioned, that atmospheric processes and spectra of QBO, if doubled, will remind the spectra of Polar motion at annual and Chandler frequency. So, his works should be referred to. In my opinion, the hypothesis of Chandler wobble, El Nino and QBO teleconnections is not proved, but is interesting. Yu. Avsyuk, in particular, pointed out, that full moon in Perigee happens every 412 days, what is somewhere close to the Chandler frequency. Chandler spectrum pike is splitted and should be forced by something, otherwise it would damp. But the frequencies do not match exactly. Tidal effects for LOD are well modeled, but even here we do not know exactly, why long-term, 20 and 80 – year modulations exist. The working hypothesis is angular momentum exchange in the Earth interior. Thus, there are unresolved problems in geophysics, including mentioned in the article, and we need ways to solve them. At that, the possible synchronizations of geophysical and astronomical processes look intuitively very beautiful, but need further development.
* * *

---

## Author Comment (AC3) · 5 Jan 2021

Prof. Zotov preceded by »

» To my knowledge, the problem discussed is not so easy, as it could be imagined. Of cause, El Nino oscillation and QBO seems to be reproduced by GCMs, but predictions are still comprehensive, for example strong El Nino 2016 was unexpected. At the same time LOD started to decrease after 2016 and it was also unexpected.

This is at best a 2nd-order effect. The consistent cycles of LOD are thoroughly explained by periodic tidal forcing. Figure 1 below shows how well the tidal factors model the dLOD/dt, where it also clearly reveals the repeating 18.6 year nodal envelope over the 3 panels.

[Figure]

» Very strong El Nino repeats at the beginning of the declining phase of LOD, what makes us to think about possible connection of these phenomena.

Yes, there may be some 2nd-order effects here, but the 1st-order mechanism must be explained first.

» But simple tides also do not explain what we observe.

I suggest they do, at least in that they may explain the 1st-order observations. It is entirely possible that some other mechanisms can explain the 2nd-order variations.

» Chandler wobble frequency does not present anywhere else in Earth systems, thus we do not see any pike in the excitation spectra at 0.843 cpy, at the same time we know from theory, that Chandler frequency is somewhere near resonance.

Like the QBO, the Chandler wobble is essentially a wavenumber=0 mechanism, which can only occur for behaviors that have no longitudinal dependence. In other words, across longitudes any axial torque applied is invariant to the inertial moment response. Thus, only the draconic/nodal 27.2122 day lunar cycle can provide a 1st-order forcing stimulus. In contrast, the LOD response must include the variation of land relief and tidal bulges which interact strongly with the tropical/synodic 27.3216 day cycle. That is also why the higher wavenumber ENSO (which is confined to the Pacific ocean) is much more sensitive to the tropical cycle than the draconic cycle.

» Amplitude of Chandler wobble is changing with 40 and 80-year modulations. In the combinations of solar and lunar frequencies we can find some close-bye modes.

This is a 2nd-order effect that may be more adequately explained after the 1st-order effects are substantiated.

» The interests of N. Sidorenkov, for sure, are in the field of interrelations of frequencies of El Nino, QBO and Chandler wobble. And in his book and papers this question was risen decades ago.

[Figure]

Yes indeed, I transitively referenced Sidorenkov in a citation and in a previous response to this submission.

» I did not quite understood the author's calculations, why 0.42 – the remainder of the division of solar year by draconic month should be doubled to satisfy 0.84 cpy frequency of Chandler wobble, but I know that in the publications of Sidorenkov it is mentioned, that atmospheric processes and spectra of QBO, if doubled, will remind the spectra of Polar motion at annual and Chandler frequency. So, his works should be referred to.

The 0.42y period is a sample-and-hold behavior of QBO generated by modulating the draconic (or nodal 27.212 day cycle) lunar forcing with an annual impulse generating a lagged response. As described in Mathematical Geoenergy (Wiley/2018), it should have the predicted frequency response peaks as shown in Figure 2.

The 2nd, 3rd, and 4th peaks listed (at 2.423, 1.423, and 0.423) are readily observed in the power spectra of the QBO time-series. When the spectra are averaged over each of the time series, the precisely matched peaks emerge more cleanly above the red noise envelope — see the bottom panel in Figure 3.

The inset shows what these harmonics provide — essentially the jagged stairstep structure of the semi-annual impulse lag integrated against the draconic modulation. It is important to note that these harmonics are not the traditional harmonics of a high-Q resonance behavior, where the higher orders are integral multiples of the fundamental frequency — in this case at 0.423 cycles/year. Instead, these are clear substantiation of a forcing response that maintains the frequency spectrum of an input stimulus, thus excluding the possibility that the QBO behavior is primarily a natural resonance phenomena. This does not preclude an additional natural response that may selectively amplify parts of the frequency spectrum.

» In my opinion, the hypothesis of Chandler wobble, El Nino and QBO teleconnections is not proved, but is interesting.

Thank you for the interest.

» Yu. Avsyuk, in particular, pointed out, that full moon in Perigee happens every 412 days, what is somewhere close to the Chandler frequency. Chandler spectrum pike is splitted and should be forced by something, otherwise it would damp.

This is indeed the case, as when a forced response behavior occurs, the fundamental frequency of the Chandler wobble should precisely match the input forcing period – which is what the model described in this submission accomplishes. The doubling of the frequencies from QBO to Chandler wobble are understood by considering the topologies of the physical mechanisms – the Chandler wobble is largly hemispherically symmetric while the QBO likely shows a topological boundary due to the equatorial waveguide, inducing the reversal of the stratospheric winds. That can also be clearly observed with the semi-annual (SAO) that occurs above the equator. See Figure 4.

» But the frequencies do not match exactly. Tidal effects for LOD are well modeled, but even here we do not know exactly, why long-term, 20 and 80 – year modulations exist.

Perhaps that can be better explained after establishing a firm foundation for the collective behavior across several phenomena.

» The working hypothesis is angular momentum exchange in the Earth interior. Thus, there are unresolved problems in geophysics, including mentioned in the article, and we need ways to solve them.

That is a possibility of course.

» At that, the possible synchronizations of geophysical and astronomical processes look intuitively very beautiful, but need further development.

That will definitely occur.
* * *
[Figure]

**Fig. 1.** Measure and model of dLOD/dt showing strong tidal forcing

**Table 11.1** Physically Aliased Frequencies and Periods
of the Draconic Lunar Month Against an Annual Cycle.

| Aliased Harmonic | Frequency (1/year) | Period (year) |
|---|---|---|
| Y/27.212-10 | 3.423 | 0.292 |
| Y/27.212-11 | 2.423 | 0.413 |
| Y/27.212-12 | 1.423 | 0.703 |
| Y/27.212-13 | 0.423 | 2.363 |
| Y/27.212-14 | −0.577 | −1.734 |
| Y/27.212-15 | −1.577 | −0.634 |
| Y/27.212-16 | −2.577 | −0.388 |

**Fig. 2.** Predicted physically aliased draconic harmonics for QBO, from Mathematical Geoenergy (Wiley, 2018)

[Figure]

[Figure]

**Fig. 3.** Identified aliased draconic harmonics in QBO spectra, at 0.42, 1.42, 2.42/yr

[Figure]

**Fig. 4.** Generation of forced response dependent on selection of impulse and tidal forcing

---

## Author Comment (AC4) · 6 Jan 2021

Similar to the QBO, specific aliased harmonics should be observed in the power spectrum of the Chandler wobble if it is forced by impulse modulated tidal period. Because the $\sim$0.843/yr = 365.242/(27.2122/2) - integer(365.242/(27.2122/2)) main frequency is identified, the other sideband at 0.157 = 1-0.843 should also exist. It does appear as highlighted in Figure 1 below, both in the model and in the spectrum of the Chandler wobble time series data. A natural resonance would not produce these frequencies, only a forced behavior could plausibly match so closely.

[Figure]

**Fig. 1.**

---

## Referee Comment (RC3) · Anonymous Referee #3 · 12 Jan 2021

I addition to comments, posted in the discussion, I would like to say, that the article can be published, if works of N. Sidorenkov are referred to.

I would also recommend the references to I. Serykh and D. Sonechkin, who are trying to connect Chandler wobble, QBO and El Nino for a long period of time already. But it is not obvious, how 1.2-year period of the first one can be doubled in QBO and in El Nino further. In the theory of chaos the periods can be reduced by non-linearity, but not easily increase.

It is also interesting, that La Nina in 2020 happened, when Earth rotates very quickly.

---

## Author Comment (AC5) · 12 Jan 2021

Definitely can include reference to Siderenkov, as his book "The Interaction Between Earth's Rotation and Geophysical Processes" published in 2009 is quite comprehensive.

» "I would also recommend the references to I. Serykh and D. Sonechkin, who are trying to connect Chandler wobble, QBO and El Nino for a long period of time already. But it is not obvious, how 1.2-year period of the first one can be doubled in QBO and in El Nino further"

Given that it is straightforward Newtonian 3rd-law physics to attach the nodal cycle of the moon to the 433 day Chandler wobble cycle, the doubling of that period for the

[Figure]

QBO does seem counter-intuitive. But one must also consider how non-intuitive the semi-annual cycle of the stratospheric winds (SAO) that occurs in altitudes above the QBO must also seem. The SAO having a semi-annual cycle means that the equatorial winds reverse for every nodal crossing of the sun over the equatorial plane, so that it will reverse on a South-to-North crossing and then the next North-to-South crossing. So when this is applied to the aliased draconic lunar nodal crossings, the math works out that the synchronization period doubles to 2*433 days or ∼28 months. This may seem additionally counter-intuitive that it is *the period* and not *the frequency* that doubles, but that is just due to the precise aliasing of the faster lunar cycles against that of the slower annual cycle. In other words, where the aliasing occurs is in a sense arbitrary – it could be faster OR slower. I supplied the charts for this on a previous response.

For El Nino and ENSO, the explanation becomes more complex, as the non-linearity of the solution to Laplace's Tidal Equation will generate many harmonics that will populate the frequency spectrum, as frequency doubling and Double-Sideband Suppressed-Carrier Modulation (DSCM) of the annual impulse will densely populate the power spectrum. For example, with ENSO, the aliasing of the strong 9-day Mt tide against the annual cycle will lead to modulation at periods upwards of 100 years, and the closeness of the aliasing of the strongest fortnightly Mf tide (producing 3.8 year cycle) and the second strongest monthly Mm tide to the annual cycle (producing a 3.9 year cycle), will also produce long-term variations. See attached Figures 1 and 2.

Thank you for the review, and certainly agree that Siderenkov, Serykh, and Sonechkin along with Zotov have been working this angle for years, but the additional novel mathematical analysis is needed to make it a quantitative instead of a qualitative hand-waving model of the geophysics and geophysical fluid dynamics.

———————————————————

[Figure]

Fig. 1. A 3.8 year cycle agrees with the ENSO model driven by the fortnightly tropical cycle (13.66 days) interacting with an annual cycle, which is indicated in the middle right pane in the figur

[Figure]

**Fig. 2.** Slight difference in the Mf, Mm, and Mm tidal forcing strength can explain differences between ENSO/SOI and AMO

---

## Short Comment (SC2) · 13 Jan 2021

Power spectra of ENSO reveal numerous spectral density peaks at the periods, which are sub- and superharmonics of three different external climate system forcings with seemingly incommensurate periods (Serykh and Sonechkin, 2019). These forces are: Chandler wobble in the Earth's pole motion ($\sim$1.2 year period), the Luni-Solar nutation of the Earth's rotation axis ($\sim$18.6 year period), and the $\sim$11.5-year Sun-spot cycle.

It is shown that the best of the CMIP5-models reproduce the ENSO spatial structure, and a nonsmooth character of its power spectra more or less well (Serykh et al., 2019). However, the periods of the modeled spectral density peaks are localized at different combinational harmonics of the annual period, but not at the afore-mentioned harmonics of the three more external periodicities. Therefore, one may conclude that just the difference between peak positions in real and modeled power spectra of ENSO is the reason why the present-day forecasting models are not capable to predict El Niño with rather long lead time.

Specific characters of the ENSO autocorrelation function decreases as well as specific relationships between the spectral peak amplitudes and their serial numbers give grounds to consider the El Niño dynamics as a manifestation of the so-called strange nonchaotic attractor (SNA) well-known in the mathematical dynamical system theory (Serykh et al., 2019). This circumstance admits to believe that El Niño is predictable with no limit, in principle.

The predictability of El Niño and La Niña is investigated (Serykh and Sonechkin, 2020a). In this case, the recently discovered so-called Global Atmospheric Oscillation (GAO) is considered (Serykh et al., 2019). Assuming GAO to be the main mode of short-term climatic variability, this study defines an index that characterizes the dynamics and relationships of the extratropical components of the GAO and ENSO. Due to the general propagation of the GAO's spatial structure from west to east, another index – predictor of ENSO is defined. The cross-wavelet analysis between both of these indices and the Oceanic Niño Index (ONI) is performed. This analysis reveals a range of timescales within which the closest relationship between the GAO and ONI takes place. Using this relationship, it is possible to predict El Niño and La Niña with a lead-time of approximately 12 months (Serykh and Sonechkin, 2020a).

Using data on the distribution of temperatures in the Pacific, Indian, and Atlantic Oceans, large-scale structures of spatial and temporal variations of these temperatures are investigated (Serykh and Sonechkin, 2020b). A structure is found which is almost identical to the spatial and temporal sea surface temperature (SST) structure that is characteristic of the GAO. Variations in water temperature in a near-equatorial zone of the Pacific Ocean at depths up to about 150 meters behave themselves in the same way as variations in sea surface height and SST. At even greater depths,

variations in water temperature reveal a "striped" structure, which is, however, overall similar to that of SST variations. Variations of water temperature at depths in all three oceans spread from east to west along the equator with a period of 14 months. This makes it possible to think that the dynamics of these temperatures are controlled by the so-called Pole tides. The surface North Pacific Pole Tide was found previously responsible for excitation of El Niño (Serykh and Sonechkin, 2019). The deep Pole tides in the Southern Atlantic and Southern Indian Ocean appear to be triggers of the Atlantic El Niño and Indian Ocean Dipole (IOD). Thus, IOD manifests itself at the depth of the thermocline more clearly than on the surface of the Indian Ocean. The out-of-phase behavior of El Niño and IOD is explained by the 180-degree difference in the longitudes of these phenomena.

References

1. Serykh I.V., Sonechkin D.M. Nonchaotic and globally synchronized short-term climatic variations and their origin // Theoretical and Applied Climatology. 2019. Vol. 137. No. 3-4. pp 2639–2656. https://doi.org/10.1007/s00704-018-02761-0

2. Serykh I.V., Sonechkin D.M., Byshev V.I., Neiman V.G., Romanov Yu.A. Global Atmospheric Oscillation: An Integrity of ENSO and Extratropical Teleconnections // Pure and Applied Geophysics. 2019. Vol. 176. pp 3737–3755. https://doi.org/10.1007/s00024-019-02182-8

3. Serykh I.V., Sonechkin D.M. El Niño forecasting based on the global atmospheric oscillation // International Journal of Climatology. 2020a. https://doi.org/10.1002/joc.6488

4. Serykh I.V., Sonechkin D.M. Interrelations between temperature variations in oceanic depths and the Global atmospheric oscillation // Pure and Applied Geophysics. 2020b. Vol. 177. pp 5951–5967. https://doi.org/10.1007/s00024-020-02615-9

---

## Author Comment (AC6) · 13 Jan 2021

» "Power spectra of ENSO reveal numerous spectral density peaks at the periods, which are sub- and superharmonics of three different external climate system forcings with seemingly incommensurate periods (Serykh and Sonechkin, 2019). These forces are: Chandler wobble in the Earth's pole motion (âĹij1.2 year period), the Luni-Solar nutation of the Earth's rotation axis (âĹij18.6 year period), and the âĹij11.5-year Sunspot cycle."

The Fourier series of strongly modulated cyclic behavior is not always straightforward. Consider that ENSO is well-known to be reinforced by a strong annual impulse – the so-called spring barrier. This annual impulse when modulated by another function (such

as tidal forcing) will create a completely different set of incommensurate periods which are not sub- or superharmonics of a fundamental frequency, but as satellite peaks of the harmonics of the annual frequency. One can verify that this is occurring by applying a mirror-symmetry pattern matching of ENSO frequencies from 0 to 0.5 cycles/year to the reversed 1 to 0.5 cycles/year – see FIGURE 1 and FIGURE 2. This satellite pairing is referred to as double-sideband suppressed carrier modulation, and one of the "appropriate signal processing techniques" I refer to in the abstract, well-known by DSP/Fourier analysts.

It is crucial that any analysis of ENSO considers this pattern, as otherwise the conventional harmonic analysis will lead one to incorrect fundamental forcing periods. Unfortunately, there are no references in the literature to any ENSO studies that apply the double-sideband suppressed carrier modulation analysis. The idea suggested here is to continue along this track and further apply the nonlinear Laplace's Tidal Equation solutions that I described in Mathematical Geoenergy (Wiley,2020) which will accurately identify each of the lunisolar factors that are involved in the geophysical fluid dynamics forcing.

Thanks to Serykh, Sonechkin and others for pointing to lunisolar factors as a mechanism, but they fall short in providing the complete picture that I am suggesting should be applied, which is the motivation for submitting a more comprehensive approach (i.e. as from the abstract : "simplifying the math of fluid dynamics and applying the appropriate signal processing techniques") as an ESD Idea.
* * *
[Figure]

**Fig. 1.** Lower x-axis is the lower sideband interval (blue) and upper x-axis is the symmetric upper sideband interval (red) shown in reverse

[Figure]

**Fig. 2.** Mirror folding of the frequency axis, from P. Pukite. "Nonlinear Differential Equations with external forcing." ICLR 2020 Workshop on Integration of Deep Neural Models and Differential Equations. 2020

---

## Author Comment (AC7) · 12 Feb 2021

My earlier response got detached from the RC2 comment, link below

https://editor.copernicus.org/index.php/esd-2020-74-AC2.pdf?_mdl=msover_md&_jrl=430&_lcm=oc108lcm109w&_acm=g

[Figure]

---

## Author Comment (AC8) · 12 Feb 2021

Thanks for the comments, with summary response below.

As the comments were mainly directed to specific points (which were addressed in individual responses) one must also consider the commonality and parsimony of the proposed model. So my first response to the comment by reviewer RC1, which was mainly focused on ENSO, did not consider the comprehensiveness of the tidal model, which applies a common-mode forcing scheme across the various observed behaviors. In particular, besides QBO and the Chandler Wobble, the constraint of the well-understood length-of-day (LOD) variation plays a significant role in reducing the degrees of freedom allowed when matching to the observations. Figure 1 shows an

[Figure]

ENSO fit while simultaneously maintaining a close match of the ENSO forcing input to that used to model the Earth's dLOD, which is also tidally driven. Figure 2 shows the calibration of the ENSO tidal factor input to that estimated by Ding and Chao – notice how well the strongest tidal forcing factors align in strength. The difference is that a solid body inertial response (i.e. LOD rotation speed variations) is linear, while the ocean's response is nonlinear due to the geophysical fluid dynamics of a basin (i.e. ENSO thermocline sloshing).

So the comprehensiveness of the proposed approach has merit in interpreting a range of geophysical behaviors, making it a candidate for further consideration.

REFERENCES

[1] H. Ding and B. F. Chao, "Application of stabilized AR‐z spectrum in harmonic analysis for geophysics," Journal of Geophysical Research: Solid Earth, vol. 123, no. 9, pp. 8249–8259, 2018.
* * *
[Figure]

[Figure]

**Fig. 1.** Lower panel shows model fit to dLOD using known tidal factors, while the upper panel shows an ENSO fit applying the LTE transfer function to the same dLOD forcing, i.e. scaled linear to nonlinear map

[Figure]

**Fig. 2.** Ding & Chao [1] apply an AR-z technique as a supplement to Fourier spectral analysis to isolate the tidal factors in dLOD. Red points used in the ENSO model align to strongest factors